# Re-Enchanting Political Theology

**Jeremy H. Kidwell** 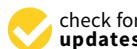

Department of Theology and Religion, University of Birmingham, Birmingham B15 2TT, UK;
j.kidwell@bham.ac.uk

**Abstract:** For this Special Issue which confronts the ways in which the question of pluralism represents both haunting and promise within modern political theology, I explore the presence of pluralism in the context of the environmental crisis and religious responses to issues such as climate change. Following Jason Ā. Josephson-Storm, I suggest that models of disenchantment are misleading—to quote Latour, "we have never been modern." In engagement with a range of neo-vitalist scholars of enchantment including Rosi Braidotti, Karen Barad, Isabelle Stengers, Jane Bennett and William Connolly, I explore the possibility of a kind of critical-theory cosmopolitics around the concept of "enchantment" as a possible site for multi-religious political theology collaborations and argue that this is a promising post-secular frame for the establishment of cosmopolitical collaborations across quite profound kinds of difference.

**Keywords:** enchantment; pluralism; political theology; political ecology; weber

## 1. Introduction

The period 2018–2019 will likely be known by historians as years in which civil disobedience and public demonstration on environmental issues became mainstream.[1] From the recurring school strikes for climate to extinction rebellion demonstrations, one surprising feature has been the visibility of faith communities and religion. Demonstrations in which I have been a participant or observer over the past five years have included a range of rituals underpinning actions including guided meditations and prayer. Groups such as Christian Climate Action are now prominent and religious leaders across a variety of faith traditions are visible among those participating in these unmitigatedly pluralistic actions. It should not come as a surprise that the religion on display is messy: traditional, implicit, hybrid, and experimental. The spiritual practices which underpin discussion and protest are eclectic and alliances are not carefully choreographed or negotiated but are, at least apparently, being formed in an *ad hoc* way. For this Special Issue which confronts the ways in which the question of pluralism represents both haunting and promise within modern political theology, I want to explore this presence of pluralism in the context of the environmental crisis and religious responses to issues such as climate change. Before considering this primary task, which will be to probe the concept of enchantment, some preliminary comments are in order as to how eco- and political- have been recently engaged.

It is important to acknowledge at the outset that ecologically specific visions are rather uncommon to the wider discourse of political theology as it has emerged in recent decades. The as-yet-unintegrated nature of ecological political theology does mean that these discussions are, at least for now, speculative exercises, which one may hope will charge the collective imagination of political theology in the years

---

[1] It is, of course, impossible to truly predict the significance of current events, but as evidence of the prominence and broad-based support for these actions one can turn to an article published in *Science* on 12 April 2019, "Concerns of young protesters are justified" which has now has more than 20,000 signatories, all scientists.

to come. Along these lines, this essay will bring together some disparate lines of inquiry in order to scope out the relationship between pluralism and political theology particularly inasmuch as it is construed as an ecologically expansive programme. In a way, the pragmatic plural spiritualities on display at contemporary demonstrations, which I have mentioned above, hint that the plurality of nature might provide fertile ground for more successful political theology in pluralistic times. Catherine Keller, also drawing on Stengers, affirms the post-propositional nature of collaboration in this space. She suggests that this is a kind of "apophatic entanglement."[2] When we settle into the pedagogies of nature and the pluralities that this entails, new possibilities for collaboration and exchange seem possible. This is not about transcending difference, but rather as Stengers proposes, a kind of provisional cosmopolitical settlement:

> In the term cosmopolitical, cosmos refers to the unknown constituted by these multiple, divergent worlds and to the articulations of which they could eventually be capable. This is opposed to the temptation of a peace intended to be final, ecumenical: a transcendent peace with the power to ask anything that diverges to recognize itself as a purely individual express of what constitutes the point of convergence of all. There is no representative of the cosmos as such: it demands nothing, allows no "and so . . . " And its question is therefore intended primarily for those who are masters of the "and so . . . ," we who with our heavy doses of "and so . . . " may well, in all good will, identify ourselves with the representatives of problems that concern everyone, whether we like it or not.[3]

In the remainder of this essay, I explore the possibility of a kind of critical-theory cosmopolitics around the concept of "enchantment" as a possible site for multi-religious political theology collaborations. Though such a suggestion might have seemed counter-intuitive in past decades, with enchantment belonging to a rather specific kind of eco-theology, there has been a resurgence and diversifying of interest in enchantment as a politically and philosophically salient concept by a range of scholars. In this essay, I pursue some of the ways that one might seek to open up political theology in dialogue with these new developments. My reflections in this essay have been provoked by the recent book by my friend and former colleague Joe Rivera, *Political Theology and Pluralism: Renewing Public Dialogue*. In his book, Rivera argues that we may want a political theology which can accommodate our increasingly pluralistic societies and he goes on to suggest that "liberalism fosters pluralism". As such, Rivera argues, liberalism might form the basis for some kind of shared democratic project: as a "sheer political mechanism, liberalism represents a pragmatic political arrangement based on a social contract laboriously forged among its citizenry."[4] Rivera is keen to suggest that secularism is not a threat to his and my tradition of Christianity. In his account, following Marcel Gauchet, secularisation is not about the forced de-Christianisation of societies, but rather secularism is "Christianity's gift to the world".[5] I agree with Rivera's characterisation. Though it has been used by hegemonic regimes as an instrument of control, Christianity is not itself hegemonic, and conversely, enchanted worlds are not always liberative. I disagree, however, with Rivera and Gauchet's suggestion that disenchantment is a natural outcome of Christianity, and that such a state of affairs might be accepted or desired. I appreciate that the kind of disenchantment that Rivera and Gauchet are arguing for parallels, to some extent, Bruno Latour's critique of modernity as consisting of a kind of intellectual purifying of categories. So, in some cases, our models of an "enchanted" world might be hermetically sealed off from less or non-immanent forms of divinity (or vice-versa) and this kind of purified account of immanent enchantment is something to be avoided.[6] This is a more benevolent

---

2  (Keller 2015, p. 193). See also (Keller 2018).
3  (Stengers 2005, p. 995).
4  (Rivera 2018, p. 16).
5  (Rivera 2018, p. 37).
6  I am grateful to Joe for clarifying this point for me in personal correspondence (25 July 2019).

account of disenchantment than (as I will explain below) the successors to Weber's account had in mind. However, I remain unconvinced that disenchantment is a pre-requisite for successful pluralistic societies, or that enchantment necessarily overlaps with divinity. It seems possible to argue that categories are rather more fluid than this in practice. This fluidity comes into sharper focus if one constructs an *ecological* political theology—that is, one that is built up from individual organisms, bodies, lives and their uneasy cohabitation on this small biosphere. As I will argue below, on this basis, we may actually find that the world has never really been disenchanted and I will go on to suggest that a new kind of political theology which builds on enchantment may offer an account of a kind of properly liberal politics in the sense that "politics" might become about the liberation of all life on earth.

## 2. Disenchantment and Political Theology: A Brief History

The notion of disenchantment has been entangled with secularisation since its earliest formulations. The classic point of reference is Keith Thomas's work, *Religion and the Decline of Magic*, but one needs to look earlier to J.G. Fraser to find the development of the three-stage theory of civilizational advance which proposed that societies progress from magical understandings to religious ones which are finally superseded by scientific understandings.[7] Fraser summarises this thesis in a letter to a fellow anthropologist in 1898:

> I am coming more and more to the conclusion that if we define religion as the propitiation of natural and supernatural powers, and magic as the coercion of them, magic has everywhere preceded religion. It is only when men find by experience that they cannot compel the higher powers to comply with their wishes, that they condescend to entreat them. In time, after long ages, they begin to realise that entreaty is also vain, and then they try compulsion again, but this time the compulsion is applied within narrower limits and in a different way from the old magical method. In short religion is replaced by science. The order of evolution, then, of human thought and practice is magic—religion—science. We in this generation live in a transition epoch between religion and science, an epoch which will last of course for many generations to come. It is for those who care for progress to aid the final triumph of science as much as they can in their day.[8]

Whereas Fraser and many of his successors saw disenchantment as a necessary stage in cultural evolution, writing on the theme just a decade later, Max Weber saw this transition as part of the rationalisation brought about by modernity. In this view, as Jenkins summarises, disenchantment is "the historical process by which the natural world and all areas of human experience become experienced and understood as less mysterious . . . conquered by and incorporated into the interpretive schema of science and rational government . . . increasingly the world becomes human-centred and the universe—only apparently paradoxically—more impersonal."[9] A key account of rationalisation and its corollary, disenchantment, can be found in Weber's 1917 lecture, "Science as a Vocation" which is concerned with "the inner attitude of the scientist himself to his profession."[10]. Inhabiting this vocation is not, Weber insists, about the actual acquisition of knowledge. To explain, he turns to an example from everyday life: "Unless we happen to be physicists, those of us who travel by streetcar have not the faintest idea how that streetcar works. Nor have we any need to know it. It is enough for us to know that we can 'count on' the behavior of the streetcar."[11] Thus, modern life is not about the possession of, but rather the (supposed) accessibility of knowledge. This sensibility has knock-on effects:

---

7   See especially (Josephson-Storm 2017, chp. 5).
8   James Frazer, *Selected Letters of Sir J. G. Frazer*, ed. Robert Ackerman (New York: Oxford University Press, 2005), pp. 137–38, cited in (Josephson-Storm 2017, p. 134).
9   (Jenkins 2000, p. 12).
10  (Weber 2004, p. 12).
11  Cf. (Weber 2019, p. 12).

Increasing intellectualisation and rationalisation does not mean an increasing general knowledge of the conditions in which we live. It means instead something else: the knowledge, or the belief, that if one only wanted, one could establish what these conditions are—that there are, in principle, no enigmatic and unpredictable forces that are here at work, but rather that all things—in principle—can be controlled through calculation.[12]

It is important to emphasise this brief clause, "in principle" as in the next sentence, we arrive at Weber's famous formulation of disenchantment. He goes on to suggest, "that [deference to calculation] in turn means the disenchantment of the world."[13]

In Weber's account, rationalisation is not just a straight-forward consequence of human progress and achievement. In actual fact, Weber's account of progress is far more ambivalent. Whilst Jenkins suggests that in Weber's disenchanted world, "everything becomes understandable and tameable, even if not, for the moment, understood and tamed", more recent interpretations have rendered a more ambiguous view.[14] Keith Tribe suggests, in contrast, that in Weber's view, human deference to science is aspirational, perhaps even tragic. One pursues this kind of comprehensive calculative knowing (e.g. science) precisely because their experience in the modern world is so fragmentary and difficult to hold together intellectually.[15] Some readers also find a kind of proto-existentialism in Weber's concern to develop a sense of vocation in the context of the world as one finds it. As Owen and Strong put it, in reference to Weber's later 1919 lecture on *Politics as a Vocation*, "political education, as Weber conceives of it, consists in being trained to accept the realities of the world in which one lives."[16] This concern animates Weber's ongoing attention to the press of external intellectual forces, such as rationalisation. Pluralism is another of these forces that persons must face, in many ways also tragically. In the 1917 vocation essay, Weber argues that there is no transcendent neutral ground on which one may seek commensurability: "life is about the incompatibility of ultimate possible attitudes and hence the inability ever to resolve the conflicts between them."[17] The tragedy may not be a permanent one and one gets the (albeit fleeting) sense that for Weber, magic is not wholly unavailable, but rather has been separated from our *most* corporate, that is societal, life. This is on display in his 1917 speech when he insists: "Precisely the ultimate and most sublime values have retreated from public life either into the transcendental realm of mystic life or into the brotherliness of direct and personal human relations."[18] It is impossible to do justice in the space of this article to Weber's views on mysticism and magic, both of which were in active development throughout his life. It is worth noting, however, that alongside Weber's strong criticism of certain kinds of mysticism, he also held out the possibility of a valorous pursuit of mysticism. As Josephson-Storm suggests, "we might imagine that for Weber, mysticism is the last route to access the transcendent God expelled by reason."[19]

As one looks forward to the development of political theology, it is important to appreciate how the reception history of Weber's narration of disenchantment shed his ambiguity and sharpened the sense of permanent anomie and alienation. To take just one example, Adorno grasps Weber's iron cage and goes on to suggest that, "The more the world is emptied of an objective meaning and the more it becomes thoroughly absorbed by our own categories and thus becomes our world, then the more we find meaning eradicated from the world; and the more we find ourselves immersed in something like a cosmic night—to express it in a modern way. The demystification (Entmagisierung)

---

[12] (Weber 2019, p. 66).
[13] (Weber 2004, p. 13). Weber introduces this concept elsewhere and earlier, as in 1905 in (Weber 1992, p. 61). For a comprehensive survey, see (Josephson-Storm 2017, chp. 10).
[14] (Jenkins 2000, p. 12).
[15] (Weber 2019, p. 66). David Owen and Tracy B. Strong also indicate the link the Nietzsche and tragedy, particularly in their introduction to the second lecture, cf., (Weber 2004, pp. xlv–l).
[16] (Weber 2004, p. xlvi).
[17] (Weber 2004, p. 27).
[18] (Weber 2004, p. 30).
[19] (Josephson-Storm 2017, p. 298).

or disenchantment of the world—to use an expression borrowed from Max Weber—is identical with a consciousness of being barred out, of a darkness in which we all move."[20]

As I have already hinted, Weber's actual relationship to the magical or enchanted world was complex, and is not necessarily represented in contemporary accounts of his theory. In particular, Jason A. Josephson-Storm's comprehensive treatment of the "myth of disenchantment," highlights Weber's interaction with contemporary German occult figures and ongoing engagement with non-western religions. However, the inter- and post-war reception of Weber's work, particularly as it comes to Carl Schmitt, tends to work with the formulation I have highlighted above. Along these lines, in its early formulation in the work of Schmitt, one finds a diagnosis of disenchantment in continuity with this (more narrow) Weberian account represented by Adorno.[21] Schmitt seeks to displace some the secularity of Weber's vision, whilst preserving the sense of the enchantment of the world and our everyday imaginaries. Along these lines, Saul Newman suggests that Schmitt "accepted the secularisation hypothesis although he rejected its liberal conclusions ... he accepted the idea that modernity is founded on a progressive secularisation of religious concepts and categories resulting in an experience of 'disenchantment'—a loss of a sacred, transcendent dimension in society."[22]

Political theology generally tends to follow this neo-Weberian framing and when enchantment does appear in contemporary political theology, it tends to be ascribed to theologies of techno-capitalism. Across these different inflections, one finds a common conviction—nested within tacit agreements about secularization—that whether one wants to celebrate or lament it, disenchantment is the reality of the day. I would tentatively suggest that this assumption of disenchantment has led, at least in part, to the marginalisation of more ecologically broad accounts in political theology to date. Within the broader orbit of religious studies and postsecular critical theory, this assumption of disenchantment has come under serious question. Bruno Latour poses this query early on in his book, *We Have Never Been Modern*. Josephson-Storm demonstrates quite comprehensively, through historical and archival work on key modern thinkers including Descartes, Bacon, Kant, Comte, Weber and Walter Benjamin, that even the champions of rationalisation, mechanism, and disenchantment have rarely held this conviction in a simple way. The suggestion that various forms of enchanted life and matter swirl around us, albeit in different configurations from one generation to the next, maps onto work in contemporary sociology of religion which has highlighted the remarkably commonplace belief in some form of paranormal phenomena (whether angels or UFOs), which transcends class, race, political affiliation and nationality.[23] As one study suggests, "The paranormal is normal ... Statistically, those who report a paranormal belief are not the oddballs; it is those who have no beliefs that are in the significant minority."[24] While one may accept that this take on enchantment is a recent and minority view, and that the supposed rupture with the past which the disenchantment thesis pre-supposes are self-reifying constructions, the legacies of disenchantment nevertheless haunts discussions of the environment in political theology. In this article, I would like to provide a constructive, if tentative, proposal. Following the work of Isabelle Stengers, mentioned above, I will suggest that the time might be ripe to test out some provisional connections across contemporary metaphysics, political philosophy and political theology in a sort of "and yet ... ".

---

[20] Theodor Adorno, *Kants Kritik der reinen Vernunft*, 1959; cited in (Josephson-Storm 2017, p. 206).

[21] See, for example, Schmitt, "Die Tyrannei der Werte," in *Der Tyrannei der Werte*, ed. Carl Schmitt et al. (Hamburg: Lutherisches Verlagshaus, 1979), p. 35. See also commentary on the relationship between Weber and Schmitt on disenchantment in (McCormick 1997).

[22] (Newman 2019, p. 24).

[23] This literature is summarised in (Josephson-Storm 2017, chp. 1).

[24] (Bader et al. 2010, p. 194).

### 3. Post-Secular Politics

Though many of the authors I survey here may not describe it in this way, it is important to underline the way that this recent turn to enchantment arises in conjunction with a postsecular moment. The confidence of mid-twentieth century theorists such as Peter Berger in the steady march of modern societies towards secularity has been firmly displaced as those same modern societies have continued to be persistently—and in some cases even increasingly religious. To be fair, religion in its postsecular forms, can often come in rather different clothing: this includes more pervasive forms of implicit religion[25], non-institutional spirituality[26] and vernacular or folk religion[27] just to name a few framings. My point here is that this confluence between interest in enchantment and postsecularity opens up an opportunity to emphasise an approach to political theology which can more readily account for the entangled and complex relationship between contemporary neoliberal citizenship and religious identity.

It is also important to appreciate the degree to which environmentalism has also recently begun a process of desecularisation. After decades of environmental science and public communication of findings, there is an emerging consensus that public response to problems like habitat loss, climate change, or mass species extinction, cannot be provoked by the dissemination of scientific studies. One can see a turn in environmental science, and particularly in the environmental values literature, towards explorations of the value of culture, aesthetics and religion.[28] Underpinning this turn is an emerging sense that there are ways of thinking and being which sit much deeper and which are embedded in cultural and religious ways of knowing which inflect, dampen and prohibit moral response to these emerging crises.

Space does not permit a full-fledged sociological exploration of the dynamics of spiritual and religious participation in popular or professional environmentalism, and many of these movements remain nascent or are so new that researchers are still in the early stages of fieldwork. However, given that I am arguing for a new kind of political theology in light of novel political circumstances, a brief description of the political valence of this spiritual and religiously particular action as it has been emerging in environmental protests and demonstrations will help to set the stage for subsequent discussion in this section. To summarise the range of possibilities: across environmentalist public demonstrations, one finds religion surfacing in three key ways: (1) in the context of organised co-located inter-faith events and blocs, (2) on the occasion of spontaneous and explicitly religious ritual acts in the midst of protest (e.g., worship, prayer, meditation, etc.) and (3) in the experience by participants in the act of protest itself as a spiritual experience. On the matter of the first category, the ubiquity of religion on display at protests in this way over the past decade was prominently launched with the People's Climate March which occurred on 21 September 2014 in New York. This action included what one of the organisers, Fletcher Harper suggests as, "10,000 of these people gathered for a three-hour, multi-faith prayer and invocation service on 58th Street between 8th and 9th Avenues in Manhattan."[29] There were a series of follow-on events, in the run-up to the Paris COP21 summit on climate change, including a "people's pilgrimage", online multi-faith storytelling about climate change organised by a group called "Our Voices", and a batch of petitions from faith groups with 1.78 million signatures presented to Christiana Figueres, Executive Secretary of the United Nations Framework Convention on Climate Change. Alongside these kinds of organised interfaith activities, there have been a variety of instances of less explicitly "denominational" activities. These forms of spiritual activism are also not limited to climate change. To give one example, I observed guided meditation sessions facilitated as part of a 2019 Extinction Rebellion demonstration in Bristol, UK. Stefan Skrimshire recounts an observation of

---

25  (Bailey 2010).
26  (Woodhead 2017).
27  (Bowman and Valk 2014).
28  See particularly (Ives and Kidwell 2019) and (Cooper et al. 2016).
29  (Harper 2016).

a similar, but more spontaneous ritual act in the context of the London Extinction Rebellion protest earlier this year on 14 April 2019.[30] He recounts a young woman who initiated a spontaneous call to prayer and observes:

> When she begins to sing the prayer, two things happen. First, several people join in with the words or begin whooping. Second, I realise that the whooping may not only be for her: the police are now moving amongst the crowd and begin arresting people at random ... Zaltash breaks off briefly to chastise the police: "we're in the middle of prayer!" When she has finished she says, "You are invited to kiss the ground and place your forehead upon it three times, if you wish." A significant number of people follow her. She finishes with "blessings to you all. You are oneness".

As Skrimshire suggests, these ritual and religious acts serve not merely as a "reflective 'aside' from the main protests" but are "acts of protest in themselves". Of course, none of these three aspects, whether formally organised interfaith activity or spontaneous ritual, are completely novel. Earlier research drew attention decades prior to the underlying spiritual aspects even in apparently secular radical environmental groups in the USA.[31] However, what seems to have decisively shifted is the ubiquity of the "religious" and the "spiritual" at these demonstrations.

It may be tempting to dismiss this context, after all, scholars in activist and festival studies have observed how these spaces are provisional and even liminal. Such liminality might have negative implications in terms of the viability of these spaces for political theology. If these bright moments of ad hoc togetherness merely dissolve as participants return to their everyday lives, one might think that the political horizon here is quite limited. There is some truth to this claim, I think, and it is important to appreciate the provisionality of these new spaces for spiritual experimentation and collaboration. However, it is important to view this new rise in religio-spiritual protest alongside the decline of formal collectivities. There are some ways in which everyday experience of political community is itself shifting quite dramatically, which makes the appearance of enchantment in these spaces of mobilisation potentially more universalisable. I am thinking, particularly here, of the work of scholars such as Robert Putnam who observed at the turn of the millennium the decline in participation in smaller collectivities and the turn towards more individualised leisure activities, in his book "Bowling Alone".[32] This has paralleled a similar and now well-known decline in the attendance of worship services.[33] My point here is that it is a mistake to juxtapose provisional political spaces with more permanent ones, as the latter seems to be in sharp decline and the former seems to be, at least for now, the preferred (or only) option. In this way, the underlying spontaneity of enchantment may prove helpful.

## 4. Seeking (Re)Enchantment

This new political liminality maps onto recent work in political theology by ecologically oriented scholars who seek to account for the political liminality of multi-species encounter. Though the context of pluralism is generally assumed to relate to nation states or major urban centres and their human inhabitants, Peter Scott has recently flipped this context and in a political theological register, suggested that we begin by recognising that "nature is a plurality" and following on from this acknowledge that "our participation in it is a site of learning."[34] His point is that typical deference to civil society as the space for forming and negotiating our conceptions of plurality (and by extension the underlying

---

[30] (Skrimshire 2019).
[31] (Taylor 1991).
[32] (Putnam 2000). One finds a similar suggestion in the work of anthropologist Daniel Miller, who, in his own way notes the decline of socio-political units beyond the household.
[33] See, for example (Heelas et al. 2005).
[34] (Scott 2015).

assumption that "nature" is a noncultural space), bypasses a more original conception of plurality which we inevitably inhabit as human animals located in particular social ecologies. In a similar way, drawing on Alfred North Whitehead and the work of Isabelle Stengers, Michael Northcott has also recently argued that "ecosystems are societies". In Northcott's conception, the metaphysics of creaturely being involves a negotiation across the boundaries of many different kinds of organisms.[35] The use of plural "ecosystems" by Northcott is important here, as it ties in with his account of "parochial ecology" in which "communities of place recover from the universalising hegemony of State and corporate actors a collective sense of responsibility for their own locale."[36] Political community, in this conception, is not composed simply of parochial *homo sapiens*, but rather involves the recognition that "the political" is inextricably entangled with "the natural" and that political community involves a lively congress of many different kinds of organisms. Common to these cosmopolitical proposals (to use a phrase by Stengers) by Scott and Northcott is an emphasis on the pre-existence and enduring character of pluralism as a reality of ecosystems and the fact that these plural societies are themselves a key layer for human politics.

These new framings of eco-pluralism also hint at the need for enchanted forms of eco-political response to the environmental crisis. To appreciate the significance of this turn, however, requires a brief return to the polemics of disenchantment. Building on my account above, it is important to appreciate how a world which has been disenchanted (or at least diagnosed as such) is not simply rendered as lacking in distributed divinity, but is as Heidegger framed it, becomes a standing reserve. This world is full of purposeless and lifeless matter which is ready to be appropriated and fashioned for any purpose that a human person might dream up. In other words, disenchantment is the sharp edge of several modern dualisms. Along these lines, the physicist and philosopher Karen Barad argues that "the inanimate-animate distinction is perhaps one of the most persistent dualisms in Western Philosophy ... It takes a radical rethinking of agency to appreciate how lively even "dead matter" can be."[37] The primary charge here is that a world full of matter which is despiritualised is rendered inanimate, and the work of re-rendering the other-than-human world as lively, and by extension, demanding our sacrificial care, requires a radical epistemological reconfiguration. The desecularisation of environmental science and management that I noted above has come as a relatively recent shift, but one which I think provides an opening for political theological attention. As I will suggest below, the work of these enchantment theorists is particularly well suited to this content because they are all working quite explicitly in conversation with contemporary science. With this in mind, I briefly explore below the work of Jane Bennett and William Connolly in order to highlight some of the features in this constellation of what I would describe as a political theology of enchantment.

One finds these newly enchanted models arising in a properly postsecular academy, as atheists, agnostics, and non-theists in particular have taken up the mantle of "enchantment". In seeking to explain why this might be the case, Rosi Braidotti suggests that in spite of its former militance, contemporary critical theory maintains a "residual spirituality".[38] Echoing Latour's point that we have never been modern, this "spirituality" comes, according to Braidotti, through political philosophical borrowing by Enlightenment political philosophers for the liberal project. Braidotti is sensitive to the political realities of postsecular pluralism, and the ideal, in her formulation (which, like Keller and Stengers, draws on Deleuze) is to take up a form of critical theory that is not tied to present conditions by an oppositional posture, but rather a reconception of the conditions for political and ethical agency might are "affirmative and geared to creating possible futures." In her view, "ethical relations create possible worlds by mobilizing resources that have been left untapped, including our desires and

---

[35] (Northcott 2013, p. 78).
[36] (Northcott 2015, p. 101). See also (Northcott 2012).
[37] (Barad 2007, p. 419)
[38] (Braidotti 2008).

imagination"[39]. For Braidotti, then, we are free to find enchantments in the context of a Deleuzeian neo-vitalist philosophy, but the overarching point here is to dismantle confidence in the existence of a univocal "western" ("occidental") intellectual project and appreciate the presence-day co-existence of "multiple modernities".[40]

Perhaps the most well-known recent attempt to recover the concept of enchantment can be found in the work of Jane Bennett with her two books: *The Enchantment of Modern Life* (2001) and *Vibrant Matter* (2010). Common to Bennett and other similar projects is an intellectual genealogy which begins with Spinoza, proceeds through Nietzsche and arrives at Deleuze, particularly in Bennett's case, with the concept of "material vitalism".[41] As Bennett herself notes, "Spinoza was not himself quite a materialist".[42] Bennett's political theory aligns with the more provisional and intermediate approaches I have highlighted above, as she avoids Marxist-style critique of hegemonic forces and national politics, and prefers to speak of "micro politics". In Bennett's conception, enchantment is an explicitly political category, though this is not quite "politics" in the sense that one might expect it. These enchanted things we encounter have the potential to "surprise, fascinate, disturb and provoke wonder" and our response to these aesthetic evocations forces us to sharpen our *relationship* with "things" whether this might be attachment or repulsion.[43] It is important to appreciate that in Bennett's project, enchantment is not restricted to nature, or even biological organisms. She is alert to the possible invocation generated by digital devices, and suggests that the human response to enchantment is not necessarily just wonder, but may be various: "the encounter with animated objects would call sometimes for demystification, sometimes for appreciation of the ability of nonhuman things to act upon us, and sometimes for both at once ... this recognition opens the way for a deliberate receptiveness toward, even an active courting of, those 'fetishes' among whose effects can be counter surprise, wonder, even enchantment."[44] The result is a calling into question the "political efficacy of critical refusal" and inviting new "modes of engagement that figure the political field as more contingent and imagine the material world as more animate."[45] One effect of Bennett's account is that the vibrant object remains a bit fuzzy (and perhaps this is unavoidable in engagement with Latourian "actants"), while the rehabilitated self is sharpened. There is an emphasis here also on that self as an *embodied* one, so enchantment is in many ways about a sensuous engagement with the world around us, as David Abram puts it: "the way the senses immediately have, of throwing themselves beyond what is immediately given, in order to make tentative contact with the other sides of things that we do not sense directly, with the hidden or invisible aspects of the sensible."[46]

William Connolly has taken up many of these themes in similar ways, and pressed them in the service of more explicitly political philosophical reflection. Drawing on complexity theory, evolutionary biology, and quantum mechanics, Connolly's wider project represents an experiment in what he calls "speculative realism".[47] A robust political pluralism has also been a persistent feature of Connolly's scholarship. Like Northcott and Keller, Connolly has also benefitted from conversation with the

---

[39]	(Braidotti 2008, p. 16).

[40]	Bruno Latour offers a similar critique in (Latour 1993).

[41]	(Bennett 2010, p. x). It is worth noting that this work of enchantment tends to be a path which forks into several different directions. Space does not permit full treatment of the full range of enchantment thinking in this paper, but some notes about the scope of work I am drawing on here are in order. In addition to the work mentioned above by Barad, Bennett and Connolly, see also resonant vitalist reflection by (Alaimo 2016, p. alaimo_2010a). A cluster of important studies can be found in Amerindian studies, particularly the recent work of (Kohn 2013). and (Descola 2013). Another major figure, Eduardo Vivieros de Castro provides a summary in (de Castro 1998). Polytheistic approaches are covered well in (Harvey 2005) and (Kadmus 2018). Another resonant but quite different cluster of work can be found in new materialist philosophy. Along these lines see (Coole and Frost 2010; Harman 2011; Meillassoux 2006; Vásquez 2011).

[42]	(Bennett 2010, p. x).

[43]	(Bennett 2001, p. 169).

[44]	(Bennett 2001, p. 127).

[45]	(Bennett 2001, p. 130). There is significant resonance here with the account of (Miller 2009).

[46]	(Abram 1997, p. 58). For development of some of these connections in a less secular direction than Bennett, see (Stengers 2012).

[47]	(Connolly 2013, p. 9).

theological metaphysics of Alfred North Whitehead. This leads him to offer an "ontocosmological" account of the earth as full of what he describes as self-organising, teleodynamic force-fields. The end of this experiment is to cultivate an awareness of the "variety of nonhuman force fields that impinge upon politico-economic life as it too impinges upon the force fields" and this awareness is intended to provide the means by which to "extend our political and cosmic sensibilities".[48] In *A World of Becoming*, Connolly argues that cosmic sensibilities lead to an awareness that the universe is full of "multiple zones of temporality, with each temporal force-field periodically encountering others as outside forces".[49] This has significant implications for human agency: events on the world stage are not merely determined by hegemonic forces, but this malevolent agency is combined with and imbricated upon by a whole host of other causal forces, many of which result from the collaborations of microscopic life. Sinking into this shared agency requires, Connolly argues, a project which can re-orient political thinking away from economic markets. As he says, these are "merely one type of imperfect self-regulating system in a cosmos composed of innumerable, interacting open systems with differential capacities of self-organisation set of different scales of time, agency, creativity, viscosity, and speed".[50]

With all of these different forces swirling around, bumping into each other, and potentially self-organising towards meaningful (if brief) equilibria and relation, Connolly advances a tentative proposal for a form of politics which might holistically address human needs and other agential forces. Echoing Bennett and Stengers, Connolly argues for the pursuit of "interim agendas" and that we should "become involved in experimental micro-politics on a variety of fronts".[51] For the purposes of this discussion, it is important to note that Connolly's composite account of temporality and his attempt to repristinate democratic grassroots politics is closely related to his account of grace, which comes mostly in the form of a critical conversation with Kant in *Fragility*. Connolly is concerned with human action and the way, even in the most benevolent configurations, it can overflow towards coercion and violence. At the same time, he is equally eager to avoid the negative, apolotical framing of the environment, common to conservation in the 20th century. This is in many ways an attempt to widen out an already liberal notion of the "citizen" through a focus on "the creative element in human life".[52] So Connolly suggests that in this account, which he calls an ethic of cultivation, "the projection of divine grace, elaborated in different ways by Augustine and Kant, is here replaced by the capacities of reflexivity and tactical work upon the self by the self. And, of course, this ethic is marked by the micropolitics by which we work on others and they upon us."[53]. But the "self" at the heart of this endeavor ultimately exhausts its own resources and we find Connolly, with his persistent intellectual honesty musing just a few pages later about the fragility of care:

> And that care, again? Where does it come from? Well, it emerges in the first instance, if and when we are lucky, from those caresses, exemplars, teachings, social connections, and shocks poured into the passive syntheses that help to compose us as human beings even before we acquire language. It is a thing of this world, passing through the portals of the sensorium to help compose relational sensibilities. It grows, along with the shocks and interruptions that disturb and spur reorientations of it, until we die or lose the fund of presumptive generosity essential to outreaching life. There is, once again, an element of *luck* folded right into the sources of ethical life; that element of luck may be located at approximately the points at which Kant invokes *grace*. Subtract the element of luck, and you are apt to end up with a morality that squeezes too much creativity from life. An ethical life needs this periodic

---

[48]　(Connolly 2013, p. 9).
[49]　(Connolly 2011, p. 7).
[50]　(Connolly 2013, p. 25).
[51]　(Connolly 2013, p. 38).
[52]　(Connolly 2013, p. 17).
[53]　(Connolly 2013, p. 128).

tension between felt, stable obligations and moments of creativity when some obligations undergo recasting.[54]

In response to this passage from Connolly, one may ask, what is luck? In Connolly's characterisation it is not quite the same as "care" which is an emergent property of outside forces and agencies which are not our own. I might tentatively say that it transcends both, lies radically outside them, but is, at least in Connolly's account absolutely necessary. In a way, "luck" is a dys-teleological way of characterising grace, which forms a part of another—albeit uniquely inorganic—field itself. This characterisation also brings us back to the theme of fragility which permeates Connolly's book and his broader analysis. The human condition is fragile, to be sure, and many recent and ancient apocalyptic narratives hover about Connolly's analysis. But fragility is given a particular kind of significance in Connolly's political philosophy as it is not exclusively something imposed from outside, it is a constituent factor of fields of force coming into interaction. They are tentative, contingent, and when seen in deep time, especially transitory.

## 5. Conclusions: Towards an Enchanted Political Theology

The new turn towards "enchantment" is by no means limited to these two accounts by Connolly or Bennett. This new field includes a wide range of religious traditions and dispositions, including Atheists, Eco-Pagans, and evangelical and mainline-protestant Christians, and in quite a number of other more hybrid configurations.[55] However, this limited account does provide sufficient orientation to these new genres of multi-species politics to support some provisional observations regarding what an enchanted political theology across this pluralistic convocation might look like in principle.

One can assume that the word enchantment will surely mean different things for different people, but the very nature of the enchanted place or creature seems to carry meanings which can only be partially grasped as an ever expanding epistemic horizon rather than a quickly foreclosed one. Here, I think that Catherine Keller's recent work in situating enchantment in an apophatic political theology has taken the discussion in precisely the right direction. Enchantment is underdetermined not because of some deliberate agenda or intellectual blurring, but rather because it draws our humanity towards a horizon which exceeds us epistemologically.

Perhaps, ironically, given this necessary fuzziness, there are also some relatively consistent features to enchanted politics, particularly in the exponents I have summarised above: (1) In some contrast to traditional radical and anarcho-marxist thinkers, such as Chantal Mouffe, relations are not configured in clearly binary oppositional ways.[56] In fact, many thinkers are working with explicitly anti-binary methodologies. (2) There is an emphasis on the individual embodied agent and their sensuous encounters with other agents, but this is a slightly different kind of liberalism, more about play and shared creativity than about less lively social contracts. An emphasis on senses and aesthetics does not depoliticise, as Stengers argues (following Abram), in actual fact our senses "are not for detached cognition, but for participation . . . "[57]. (3) This account of bodies in relation gives way to an emphasis on what might be called process, pragmatism or provisionality—a kind of necessarily experimental politics; (4) These accounts emphasise individuals, but are not atomistic. Generally, one finds an emerging account of collectives both on cosmopolitan macro-scales and micro-scales and these scales traverse individual bodies (e.g., bacteria) so scales are themselves destabilised; (5) Finally, neither transcendence or immanence are excluded. There are quite radically different accounts of what beings hover over us in these enchanted relations, but none of these accounts deny that transcendent being and immanent being are crucial aspects of human cosmopolitical life. It is on this final point that one

---

[54] (Connolly 2013, p. 132).
[55] Alongside accounts listed in the footnote above, see also (McGrath 2003 and Usher 2012 ).
[56] For an example of Mouffe's account of agonism, see (Mouffe 2005).
[57] (Stengers 2012).

finds some symmetry with Rivera's account, even if I would qualify his account of disenchantment.[58] The point here is that enchantment requires, or enables us to, hold open immanence and transcendence. What is particularly interesting to me, in light of this overarching question of pluralism is the degree to which this exercise of holding things open seems to be increasingly compatible with a pluralistic array of political philosophies. What this might mean for our attempts to re-narrate our engagement with the natural world remains to be seen. For now, let us hope that the scholars can keep up with the activists.

**Funding:** This research received no external funding.

**Acknowledgments:** Thanks are due to Michelle Bastian for first getting me thinking about the theme of enchantment in the context of a series of seminars on Strange Encounters in Deep Time, which have subsequently been written up in the journal *Environmental Humanities* 2018, vol. 10:1. I am also grateful to the conveners of the Stockholm University Doctoral School in Environmental Humanities: Christina Fredengren, Karin Dirke and Claudia Egerer and to the doctoral cohort for their very helpful engagement with an early paper I presented to this colloquia on the theme of enchantment. Thanks also to several fellow scholar-activists who have helped to illuminate my appreciation of enchantment: David Farrier, Franklin Ginn, Richard Irvine and Stefan Skrimshire.

**Conflicts of Interest:** The author declares no conflict of interest.

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
