# Peer review of "Re-Enchanting Political Theology"

_religions, doi:10.3390/rel10100550_

Round 1

Reviewer 1 Report

This is a fascinating piece of work, raising questions about the very nature of pluralism at a time when questions of how to coexist and what we coexist with are urgent. Thinking about enchantment in this setting seems apt - not least given the shape and form of certain strands of environmental protest. The impersonality of particular teleologies of disenchantment is well illustrated (e.g. via the quoting of Frazer's letter), and it is certainly an important question to ask whether such impersonality is suited for the challenge of pluralistic participation in more-than-human worlds. This having been set out nicely, I got a strong sense of the relational potential of enchantment for the challenge of thinking about and being in those more-than-human worlds.

Without wishing to be prescriptive, I did feel that while the core ideas were interesting, the impressively wide references mustered in the essay perhaps drew the argument too far away from what I would have expected to be its foundations:

1) The context of the discussion, as we are told at the start of the paper, is the proliferation of environmental protest in 2019. The author notes that they have observed such protests and their ritual underpinnings, as well as the diverse religious profiles of some of the participants. This all seems very important, both in terms of explaining why it is important to have this discussion now, and for providing evidence for the claims being made. So, put simply, why is there so little description, discussion, and reflection on the actual experience of the protests as observed by the author? It strikes me that there's a great deal to be gained by allowing the notion of enchantment to arise from what's happening in the environmental movements themselves. As it is, the experience of enchantment as it might be sensed in the protests and in the emergence of new groupings, alliances, and spiritual practices plays second fiddle to the theoretical and academic formulations of enchantment. The author clearly has a good sense of the contribution of neo-vitalist scholars, but I feel it is a missed opportunity if they don't put this scholarship more directly in dialogue with what's happening in contemporary protests and emerging movements, rather than relegating those experiences firmly to the background after the introduction.

2) It seems unfair that a paper which makes extensive use of Weberian concepts, and indeed includes Max Weber as a keyword, does not directly engage with or cite Weber's texts; especially given the recognition that we may have misconstrued many of the most taken-for-granted Weberian concepts in Anglophone circles, in light of the new translation of Economy and Society by Keith Tribe. To approach Weber through the lenses provided by Schmitt and Adorno without thinking about Weber's ideas in his own words is problematic. At any rate, it would be good to directly engage with key texts by Weber that form the basis for our understanding of his (to my mind, highly ambivalent) approach to enchantment and disenchantment; as well as The Protestant Ethic and the spirit of capitalism, I would suggest engaging with Science as a Vocation.

Author Response

I'm very grateful for this careful and gracious review of my article. I've sought to integrate all of the changes you've recommended in the revised draft which you'll find attached. In particular:

(1) Towards your first suggestion that I include commentary on contemporary protests and emerging movements, I've added a new, transitional §3 to the paper "post-secular politics" in which I try to flesh out some of the significance of these new movements for my broader argument in the paper regarding the viability of "enchantment" as a critical frame for political theology. I agree with your observation that it there should be some direct dialogue with what is "happening in contemporary protests and emerging movements" and have tried to provide a more significant (if still brief) diagnosis of what exactly seems to be going on there, and why it is significant for political theology. I haven't been able to provide a full explication of my own participant observation work, as this is still in "the field" and some months away from proper saturation and in-depth analysis. What I've tried to do here is bring on some other voices and first-hand testimony which comport with my preliminary observations. I'd be willing to omit mention of my own observation/participation in these events if this seems too unsubstantiated to include, but hope that what I've got here has struck some balance. Taking on board these suggestions has also grown the paper from 6.5k to 8.5k words, and I'm aware that much more here will also tip the balance away from the more philosophical/critical argument that I'm trying to make here regarding enchantment.

(2) Regarding mention of Weber - totally fair point and most welcome feedback. I've added a much fuller treatment here, particularly focusing on the two lectures on vocation and some relevant secondary literature which puts this all in context.

Reviewer 2 Report

Review Comments

Introduction:

Overall, writer shapes a serviceable introduction which clearly sets out the focus of the article, its main dialogue partners and theological framework. This section could however be improved in two ways. Firstly, by making the style less journalistic and conversational and  more academic in tone. A sentence at the beginning like:  ‘2018 will likely be known by historians as the year in which civil disobedience and public demonstration on environmental issues became mainstream’ provokes questions like: which historians? And why is it likely? General statements like this are better off avoided. The writing style could also be improved by using more formal language. Instead of: ‘Before I get to my primary task’ something like ‘Before considering this primary task’ might be more appropriate.  The writer should perhaps lessen the use of the first person to solidify this more formal tone. Secondly, the introduction could be made sharper and more focused if it were shorter, I suggest the writer moves some of the Scott and Northcott material to the second section. Not all key terms and concepts need to be defined in the introduction, merely sign-posted for later discussion.

Disenchantment and Political Theology: A Brief History:

This is a well-researched section, that effectively defines and puts to use the concept of enchantment. A few aspects of this section would benefit from development. The author notes:   ‘Political theology generally tends to follow this Weberian framing and when enchantment does  appear in contemporary political theology, it tends to be ascribed to theologies of techno-capitalism’. It would assist the discussion if the author gave a concrete example of this Weberian framing in the discussion. Which aspects of political theology does this framing affect? Does it impinge on Political Theology’s methodology? If so, in what way? This does not need to be a long addition, but it would improve the clarity of the discussion. Before moving on, the writer should also summarise what has been ‘banked’ in this overview discussion. What does it tell us about the areas of inquiry expressed in the introduction? The writer should gesture to things in this next section that will make sense of these observations. Such signposting will assist the reader in keeping track of the argument.

Seeking (re)enchantment

This is a rich section, possessing thoughtful digests of the relevant literature, but there are moments when the focus is on summary rather than analysis. The author mentions ‘the post-secular’, but doesn’t define explicitly what this moment is, or how it is concretely linked to enchantment. That should be the first task of section three. Again a paragraph banking the key insights from the authors surveyed should be added. By the end of reading this section, readers should know: What have these authors shown? How do these summaries assist the argument so far? The author should link this additional paragraph back to the co-ordinates set out in the introduction. It might be helpful for the author to add an additional ‘implications’ section to this article, outlining in more detail what is being shown and how it links back to the theme of enchantment.  

Enchanted Political Theology

Again, this section contains some good ideas, but the implications of them are not sufficiently interrogated. This concluding section feels half-formed and rushed. Problematically, new interlocutors are introduced in this section like Graham Harvey. Generally, it is bad academic practice to introduce new thinkers into a concluding section. The author should either delete reference to new thinkers at this late stage or add them earlier on in the discussion so justice can be done to their work and contribution. Not doing so looks a little like unnecessary name-dropping.   The key question here is: What useful work are these thinkers doing? What do they help me to say about my area of inquiry?  Overall, the conclusions of this final section seemed ill-defined and vague, in part because not all key concepts and theoretical moves were defined and contextualised beforehand. The author should state much more clearly and emphatically how enchantment impacts political theology, and what it means for theo-political praxis.

Summary Comments  

This article has some notable strengths. It possesses some rich engagements with relevant theological and political literature. It’s exploration of enchantment in relation to political theology is an intriguing and rich pairing.  It gestures at some intriguing possibilities concerning the ways in which enchantment could repair theo-political praxis.  However, the article as it stands still reads like an initial conference paper. Before publication, the author should work on the following areas:

Tighten up the writing style, using more formal turns of phrase Spend more time defining key terms and theoretical approaches Avoid excessive name-dropping. Make sure that each thinker is doing definite work in relation to the argument. Use much greater signposting to help the reader understand how the argument is progressing. Be clearer about the article’s contribution to the field. What is the key thing the reader should take away? How does this article change/progress things? What is being dismissed and what is being affirmed?

If all these areas are attended to and developed, the article would be much more focused and coherent.      

Author Response

I'm very grateful for this careful and gracious review of my article. I've sought to integrate all of the changes you've recommended in the revised draft which you'll find attached. In particular:

(1) I've gone over the language and have rendered the style a bit less conversational - and have implemented your suggestions as noted. I've swapped out first person in most instances. I would note that this is a matter of personality and style, and my attempts to 'personalise' my writing are a reflection of some deliberate writing choices based on my own non-positivistic mode of research, but I think you're right that as it was the piece was a bit too relaxed. I hope you'll find the new rendering a better balance.

On the specific claim that the past year will "likely be known by historians as years in which civil disobedience and public demonstration on environmental issues became mainstream" I've added a footnote with some evidence to back this claim. There is also now a much fuller treatment of the context of environmental activism in a new section §3 so this claim has much more context.

I found the suggestion that I shorten the introduction helpful. I've done this, precisely as advised and think the piece is better for it.

(2) For §2, Disenchantment and Political Theology: A Brief History:

I've added quite a lot more material fleshing out Weber's account of disenchantment and have changed the inflection regarding transmission of his thought. I've added a much fuller treatment here, particularly focusing on the two lectures on vocation and some relevant secondary literature which puts this all in context.

As revised, the article has grown from 6.5k words to 8.5k words, so I have focussed on revision of paragraph openings and added signposting throughout rather than adding discrete paragraphs to the end of each section. I'd be willing to add the kind of "banking" content you advise if you think this is still warranted, but do also want to maintain the pace of the writing and avoid inflating the article excessively.

(3) §3 Post-secular politics / §4 Seeking (re)enchantment

Again, I've integrated all suggestions here. This expanded to such a degree, that it became necessary to split into two sections, so you'll find there is a new expanded treatment of post-secular at the opening and more content (as also requested by another reviewer) in which I try to flesh out some of the implications of these new environmental movements for my broader argument in the paper regarding the viability of "enchantment" as a critical frame for political theology. There is some new extended commentary on what is happening in contemporary protests and emerging movements, some diagnosis of what exactly seems to be going on there, and why it is significant for political theology.

(4) §5 Conclusion: Towards an Enchanted Political Theology

In light of the expansion of sections above, I've opted to streamline this section a bit. I agree that it was problematic to introduce new interlocutors - you'll find that reference to new sources have been moved up, shifted to footnotes or dropped. Given the ways that previous content has been more explicitly signposted and drawn into sharper relief, I think that the concluding comments emerge more explicitly out of previous reflection and serve as a more obvious summary in this new draft.

Thanks again for your careful engagement with this article - I'm very grateful and will look forward to continued conversation.